# GLH: From Global to Local Gradient Attacks with High-Frequency Momentum Guidance for Object Detection

**DOI:** 10.3390/e25030461

**Published:** 2023-03-06

**Authors:** Yuling Chen, Hao Yang, Xuewei Wang, Qi Wang, Huiyu Zhou

**Affiliations:** 1The State Key Laboratory of Public Big Data and College of Computer Science and Technology, University of Guizhou, Guiyang 550025, China; 2Computer College, Weifang University of Science and Technology, Weifang 261000, China; 3The School of Informatics, University of Leicester, Leicester LE1 7RH, UK

**Keywords:** information security, artificial intelligence, adversarial attack, object detection, migration attacks

## Abstract

The adversarial attack is crucial to improving the robustness of deep learning models; they help improve the interpretability of deep learning and also increase the security of the models in real-world applications. However, existing attack algorithms mainly focus on image classification tasks, and they lack research targeting object detection. Adversarial attacks against image classification are global-based with no focus on the intrinsic features of the image. In other words, they generate perturbations that cover the whole image, and each added perturbation is quantitative and undifferentiated. In contrast, we propose a global-to-local adversarial attack based on object detection, which destroys important perceptual features of the object. More specifically, we differentially extract gradient features as a proportion of perturbation additions to generate adversarial samples, as the magnitude of the gradient is highly correlated with the model’s point of interest. In addition, we reduce unnecessary perturbations by dynamically suppressing excessive perturbations to generate high-quality adversarial samples. After that, we improve the effectiveness of the attack using the high-frequency feature gradient as a motivation to guide the next gradient attack. Numerous experiments and evaluations have demonstrated the effectiveness and superior performance of our from global to Local gradient attacks with high-frequency momentum guidance (GLH), which is more effective than previous attacks. Our generated adversarial samples also have excellent black-box attack ability.

## 1. Introduction

Information security concerns various fields, such as deep learning [1], homomorphic encryption [2], IoT [3], and others. Advancements in deep neural networks (DNNs) have fundamentally driven the application of object detection [4] in the real world, such as face recognition [5], intelligent transportation [6], industrial detection [7], and intelligent medical imaging [8]. The current deep learning-based object detection algorithms can be divided into one-stage and two-stage models. For one-stage, the model extracts feature directly to anticipate the classification and location of objects in the network, such as YOLO [9] and SSD [10]. For two-stage, the model first generates region proposals by selective search algorithm; subsequently, the samples are classified by the convolutional neural network, such as R-CNN  [11]. Although object detection is advancing by leaps and bounds, there are concerns about its security. For adversarial attacks [12], only a small perturbation needs to be added to interfere with the judgment of the model. Adversarial attacks and adversarial defenses [13] complement each other, and studying adversarial attacks not only analyzes the security of object detection models but also provides high-quality training samples for model robustness. Therefore, it is urgent to study adversarial samples for object detection.

However, there are few types of research about adversarial attacks for object detection tasks. The reason is that the adversarial attack for object detection is more complex compared to the adversarial attack for classification, which only requires a loss function. The first complexity is the gap between the datasets of object detection and image classification. Classified datasets frequently contain only a single object, whereas datasets for object detection, such as Microsoft Common Objects in Context (MS COCO) [14], usually contain multiple classes of targets in which objects cover one another and repeat perturbations can affect one another, leading to a weaker attack. The background accounts for a larger proportion compared to the target. Moreover, excessive perturbation is easily wasted on the background and increases the overhead of the attack. The second complexity is the significant structural discrepancies of the models. Concerning the output prediction of object detection, an alternative sub-optimal bounding box in the vicinity of the attacked bounding box possibly gets detected even if a bounding regression box is successfully assaulted. In addition, a large number of generated bounding boxes makes the cost of the attack more expensive and enhances the difficulty of the attack.

To solve the above problems, we propose an adversarial concerning global to local gradient attacks with high-frequency momentum guidance that focuses on object detection. Previous approaches use a norm to constrain the amount of perturbation, yet this constraint is reacting to the computer’s sensing of that perturbation. Contrast samples are meant to deceive not only the machine but also the human senses. Therefore, we compare the distortion of the picture from a global perspective and introduce SSIM [15] and PSNR metrics to determine the effect of the perturbation on the image, which better reflects the intuition of the human eyes.

From a local perspective, we differentially add perturbations for features to destroy important perceptual features of the target. The final effect is shown in Figure 1. In other words, our local perturbation effectively pinpoints the region of interest predicted by the object detection model. This is because gradients represent the feature information that the model recognizes in the object. We achieve the optimal attack perturbation by suppressing the positive gradient information. Moreover, at the same time, we introduce the Fourier transform to extract the high-frequency image of the post-attack image, which is fed into the model to obtain the high-frequency feature information. Using this high-frequency feature information as momentum to guide the next gradient attack is equivalent to correcting the direction of the attack. Our contribution points are as follows:We propose a generalization of object detection-based adversarial attacks that target images by dynamic gradient features. Our approach almost defeats the judgment of the object detection model in a white-box attack. For the black-box attack, our attack also achieves excellent results on object detection models with different structures.A local dynamic constraint module is proposed to alleviate the problem of excessive image similarity fluctuations after the attack by limiting the high-gradient perturbations, which reduces the perturbations to the background by eliminating the low-gradient information. On the whole, the similarity of the adversarial sample is improved.A momentum guidance method based on high-frequency gradient features is proposed to filter the Fourier-transformed images by high-frequency filtering. The gradient information of the processed image is added as momentum to the next iteration of the perturbation. The overall effect of the attack is improved.

In this paper, we propose a global-to-local adversarial attack based on object detection. Section 2 focuses on the related work of the thesis and the benchmark formulation. Section 3 presents the details of our proposed method. Section 4 is a detailed experimental demonstration and visualization of our proposed method. Finally, the last section is a summary and outlook.

## 2. Related Work

### 2.1. Gradient-Based Classification Attacks

Since Szegedy et al. [12] proved the existence of adversarial examples, several adversarial attack algorithms have been induced to study the weaknesses of neural networks. The study of the classification model’s adversarial attacks was also pioneered and the BFGS attack was proposed. Goodfellow et al. [16] found that the gradient information can be used as a guide for the attack and created the FGSM adversarial attack method. Although FGSM found the direction of the attack, it was often extremely troublesome to design the size of the perturbation. Therefore, the iterative version of I-FGSM [17] was worked out afterward. By iterating the subtle perturbations many times, it was easier to find the optimal perturbation value. However, this method was not as strong as FGSM in terms of migration capability and had significant limitations. The project gradient descent attack (PGD) [18], on the other hand, added a random initial perturbation to I-FGSM to avoid encountering the saddle point problem. Naturally, this perturbation was also bounded by the paradigm and had the same problem of weak mobility. Huang et al. [19] used feature differences to improve antagonistic sample mobility. Although gradient attack has been a great success in the field of classification, the application of this method to object detection has not been effective.

### 2.2. Query-Based Classification Adversarial Attack

The query-based black box attack relies only on the predicted scores to estimate the prediction of the gradient, which therefore requires multiple queries to estimate the approximate result. Ru et al. [20] used Bayesian optimization to find successful perturbations with high query efficiency by selecting the best dimensionality reduction angle of the search space for the attack. Du et al. [21] employed meta-learning to approximate the gradient estimation, which greatly reduced the number of queries required. However, all these methods required estimating gradient information. Moon et al. [22] proposed an efficient discrete substitution method to optimize query consumption. Chen et al. [23] proposed using meta-learning to reduce the number of black-box attack queries. Furthermore, this method was without computing the gradient. But these black box attack methods required certain query information and were not powerful enough for the effect of the attack.

### 2.3. Patch-Based Object Detection Adversarial Attack

Brown et al. [24] first proposed an adversarial patch for object detection by training the patch to make the classification output of the model wrong. However, the patch only focused on the classifier. In addition, this method had a significant impact on the image. In contrast, Liu et al. [25] performed the attack by adding the patch in the upper left corner of the image. The backpropagation during training only updated the patch and disabled the detection frame. The method performance has not been excellent in recent models. Lee et al. [26] improved on the former by adding an adversarial patch preprocessing to focus the model’s attention on the adversarial patch. Nonetheless, the method relied too much on the model structure. Thys et al. [27] used 2D printing techniques to hide humans from the detection system. Yet, the method was weakly generalized and allowed attacks only against a single target. Hu et al. [28] proposed AdvTexture, based on previous research, which used wearable clothes to evade the detection of multi-angle attacks in the physical world. None of the above methods provides an interpretable basis for object detection and cannot improve the model’s robustness to provide adversarial samples.

### 2.4. White-Box Based Object Detection Adversarial Attack

Existing object detection white-box counterattack methods mainly implement attacks by changing the classification loss. DAG [29] and CAP [30] implement attacks mainly by spoofing the RPN network of two-stage object detection models. To achieve migration, UEA [31] and TOG [32] attack both one-stage and two-stage detectors with metastable adversarial perturbations. Nevertheless, the above method has a weak migration.

Selection loss for object detection is composed of three components: confidence, bounding regression box, and classification. Thus the adversarial attack based on white-box object detection is also based on three predictions to generate adversarial samples.

The first part is the confidence loss of object detection. The attack confidence loss allows either adding false targets or hiding real objectives and is formulated as follows:(1)$$L_{obj}=\sum_{i=1}^{s}\left[O\ell_{BCE}\left(1,C_{i}\right)+\left(1-O\right)\ell_{BCE}\left(0,C_{i}\right)\right],$$
where $C_{i}$ represents the model’s confidence prediction output; O represents the accuracy of detecting the corresponding object; and $\ell_{BCE}$ stands for binary cross entropy.

The second part is the loss function of the bounding regression box. Attacking this loss function allows the prediction box of object detection to move away from the target, which means that the predicted and actual errors should be amplified. The equation is as follows:(2)$$\begin{array}{c} L_{bbox}=\sum_{i=1}^{s}[\ell_{SE}\left(t_{x}^{i}-{\hat{g}}_{x}^{i}\right)+\ell_{SE}\left(t_{y}^{i}-{\hat{g}}_{y}^{i}\right)\\ +\ell_{SE}\left(\sqrt{t_{w}^{i}}-\sqrt{{\hat{g}}_{w}^{i}}\right)+\ell_{SE}\left(\sqrt{t_{h}^{i}}-\sqrt{{\hat{g}}_{h}^{i}}\right)], \end{array}$$
where $t_{x}^{i}$ denotes the x-coordinate of the center of the bounded regression box predicted by the model. In addition, $t_{y}^{i}$ denotes the y-coordinate, and $t_{w}^{i}$ denotes the width value of the predicted bounded regression box. Moreover, $t_{h}^{i}$ denotes the height of the predicted bounded regression box, and ${\hat{g}}_{x}^{i}$,${\hat{g}}_{y}^{i}$,${\hat{g}}_{w}^{i}$,${\hat{g}}_{h}^{i}$ indicate the coordinates of the real label. Finally, $\ell_{SE}$ represents the sum of squared errors.

This is concluded by the classification loss of the objectives. Attacking the classification loss function then misleads the model to misclassify the target. The formula is shown below:(3)$$L_{cls}=\sum_{i=1}^{s}O\sum_{c=1}^{k}\ell_{BCE}\left(p_{i}^{c},{\hat{p}}_{i}^{c}\right),$$
where $p_{i}^{c}$ represents the classification information predicted by the model; ${\hat{p}}_{i}^{c}$ represents the real target classification information; *k* represents the total *k* predicted categories; and *s* indicates that the image has *s* detected targets. However, the above methods are all attack methods derived from classification adversarial attacks and do not target the features of the object detection dataset.

## 3. Methods

### 3.1. Overall Framework

An overall structure of our proposed GLH method is shown in Figure 2. The algorithm flow is shown in Algorithm 1.
**Algorithm 1** GLH.**Input:** clean samples $I_{C}$, perturbation value λ,μ, number of iterations T, $F^{\prime }\left(.\right)$ indicates that high-frequency image information was acquired using the Fourier transform.**Output:** 
Adversarial samples $x_{adv}$1:$\rho_{hfb}=0$2:$I_{A}^{i}=I_{C}$3:**for** 
i=0→T−1 
**do**4:   $\rho_{lfa}=Hardshrink\left(D_{clip}\left(\nabla \frac{L_{sum}}{S}\right),\lambda \right)$5:   $I_{A}^{i+1}=I_{A}^{i}+\mu \left(\rho_{lfa}+\rho_{hfb}\right)$6:   $\rho_{hfb}=\epsilon *tanh\left(\nabla \frac{L_{sum}}{F^{\prime }\left(I_{A}^{i+1}\right)}\right)$7:   $x_{adv}=I_{A}^{i}$8:**end for**9:**return** $x_{adv}$

The section in blue shows the single perturbation generation process. Before feeding the model into the image, the padding operation is first performed to complement to 640 × 640 size, because YOLOR [33] performs data enhancement for data with inconsistent image size. One of the scaled image sizes will be interpolated, thereby the generated adversarial samples will be affected by the interpolation to reduce the attack effect. Subsequently, we input the processed images into the model to get three predictions, in which we calculate the global gradient information we need by the three corresponding loss functions. By the variance AGA_Grad of this global gradient information, we selectively generate perturbations. To generate high-quality samples with dynamic constraints, we obtain local perturbation LFA_Eps with the original image after the previous padding, which is performed by adding an operation to obtain the attached image. Eventually, inverse padding is performed to obtain the original size image.

The green part is the high-frequency momentum guidance module. We get the confrontation sample for Fourier change and input the high-frequency image into the model to get the corresponding loss function. We calculate the high-frequency perturbation as the momentum guidance for the next perturbation.

### 3.2. Adaptive Gradient Attack

With image classification tasks, images in datasets are almost exclusively of a single class (e.g., CIFAR-10, ImageNet [34], and ILSVRC [35]). The proportion of the target is high, therefore the adversarial sample generation for classification often gets the direction of the perturbation through the gradient. That is, by superimposing the same perturbation for the direction of the gradient, the gradient of the image can be moved away from the normal range. In object detection, the datasets contain many objects with random size and distribution, for which the perturbation addition method of image classification attack cannot perform satisfactorily on the object detection task.

Therefore, we design an adaptive gradient attack to perform specific perturbations for different targets of different images. We find that the gradient information represents the region of interest of the model. Our proposed adaptive gradient attack (AGA) method uses the gradient information obtained from each iteration as a quantifier of the perturbation. The generalized equation of the method is shown as follows:(4)$$I_{n}^{i+1}=Clip_{\left(0,1\right)}\left\{I_{n}^{i}+\epsilon \nabla \frac{L_{sum}}{S}\right\},\mathrm{Min}\left(A\sum_{n=0}^{N}I_{n}^{i+1}\right).$$

Because the number of targets in the same image is different, we ask for the average loss of the number of targets s to obtain the gradient information. Furthermore, the size of this gradient can be used as the scale of our perturbation to control the overall perturbation size by ϵ; A represents the AP metric of the computed image. In addition, our goal is to reduce the AP value of the N samples in the datasets as much as possible while reducing the image corruption as much as possible.

For the weights assigned to the three loss functions of $L_{sum}$, we also adjusted them by adding the hyperparameters α, β, and γ. The formula is shown next:(5)$$L_{sum}=\alpha L_{obj}+\beta L_{box}+\gamma L_{cls},$$

The value of $L_{sum}$ was also adjusted as follows: the hyperparameters α, β, and γ were added. Because we are based on the white-box attack of YOLOR, we take the preset hyperparameters of the model: α is taken as 0.7, β is taken as 0.05, and γ is taken as 0.3. To be fair in the experimental setting, the experiments we compare are all with the same parameters. In the subsequent ablation experiments, a comparison of the values of these parameters will be made.

Simultaneously, we also visualize and compare our proposed AGA perturbation with the perturbation generated by I-FGSM. To ensure that the visualization is more obvious, we do not put a constraint on the perturbation for ϵ. As shown in Figure 3, it is evident that our proposed method is more target-focused and suitable for object detection attacks.

### 3.3. Local Gradient Feature Attack

The established object detection adversarial method is iteratively attacked, and we obtain unusually powerful attack results. However, we find that, if we go unconstrained in using the gradient information, although we obtain amazing results in the attack, a part of the sample perturbation drastically affects the image. The similarity of the attacked image fluctuates dramatically compared to the original image, which is detrimental to our work. This is explained by the fact that too numerously perturbed images cannot be used as suitable adversarial samples to train the model robustness. Moreover, this adversarial sample does not prove that the pixel point we attack is the knowledge learned by the model. Therefore, we introduce a dynamic constraint module to limit the perturbations and constrain the extremes of the gradient according to the number of iterations, which substantially improves the stability of the adversarial sample similarity after discarding the excessive attack perturbations. It is also proven in our subsequent ablation experiments. The equation of our perturbation generation is shown as follows:(6)$$\rho =\epsilon Hardshrink(D_{clip}\left(\nabla \frac{L_{sum}}{S}\right),\lambda ),$$
where
(7)$$D_{clip}\left(x\right)=\left\{\begin{array}{cc} x, & x<=\delta +\sqrt{\delta *(N+1)};\\ 0, & x>=\delta +\sqrt{\delta *(N+1)}. \end{array}\right.$$
(8)$$Hardshrink\left(x,\lambda \right)=\left\{\begin{array}{cc} x, & x>\lambda ;\\ x, & x<-\lambda ;\\ 0, & otherwise. \end{array}\right.$$

In addition, we define how the perturbation ρ is generated, as shown in Equation (Equation 6). Moreover, ∇ represents the acquisition of gradient information. We use $D_{clip}$ to constrain the upper limit of the gradient, as in Equation (Equation 7), as well as every time the perturbation reaches the limit, we set it to (0, 1), because too high a perturbation will destroy the image. In addition, it is considered that information with a low gradient is of little help to the prediction of the model. Therefore, it is not necessary to design perturbations based on excessively small gradients. Instead, we use the Hardshirk activation function to suppress the information with relatively low gradient information, as in Formulation (8). For the value of δ, we use a hyperparameter of 50, and N represents the number of iterations. The idea is to use an upper bound on this perturbation that increases with the number of iterations, but to constrain the perturbation by limiting the growth rate at each iteration.

Of course, a single attack cannot find the optimal attack direction. Therefore, we need to iterate over each attack to generate the best adversarial sample. The formula is as follows:(9)$$I_{n}^{i+1}=Clip_{\left(0,1\right)}\left\{I_{n}^{i}+\rho \right\},\mathrm{Min}\left(A\sum_{n=0}^{N}I_{n}^{i+1}\right).$$

Therefore, we attack by iterating the attack steps; $I_{n}^{i}$ stands for the last image, and ρ is the currently computed perturbation. As the range of pixel values after the image normalization process is (0,1), we finally constrain the image to a normal range using the Clip function.

The visual comparison of the method is also performed, as shown in Figure 4: it can be seen that our method is more focused on the object and less perturbing to the background.

### 3.4. Fourier High-Frequency Momentum Guidance

It was discovered that the high-frequency information of an image represents the semantic information of that image. Additionally, even if we remove the low-frequency texture information, the model can still detect the target of that image normally. Therefore, we design a high-frequency gradient bootstrap to reinforce the gradient attack. The specific idea is that we save the high-frequency feature gradient of the attack image after the first iteration of the attack, as well as add a perturbation of the high-frequency feature gradient to guide the image to change in the next step of adding perturbation to the attack. The formula of this high-frequency feature gradient can be expressed as the following equation:(10)$$F\left(u,v\right)=\int_{-\infty }^{+\infty }\int_{-\infty }^{+\infty }f\left(x,y\right)e^{-j2\pi (ux+vy)}dxdy,$$
(11)$$F^{\prime }\left(u,v\right)=F\left(u,v\right)*C\left(x,y\right),$$
where
(12)$$C\left(x,y\right)=\left\{\begin{array}{cc} 0, & 270<x<370,270<y<370;\\ 1, & else. \end{array}\right.$$

For the image f(x,y) after our first attack, we obtain its frequency domain image F(u,v) by Fourier variation; j represents the imaginary part unit. We get its high-frequency frequency domain image $F^{\prime }\left(u,v\right)$ by setting the high-pass filter C(x,y). In addition, for the area of the high-frequency filter, we choose 1000, due to the image size being fixed to 640 × 640, so the range of the central filter is 270 to 370. We get its characteristic image by reducing it. We then get its high-frequency time domain image $F^{\prime }\left(x,y\right)$ by inverse Fourier transform. The contour features can be seen clearly. The inverse Fourier formula is shown below:(13)$$F^{\prime }\left(x,y\right)=\int_{-\infty }^{+\infty }\int_{-\infty }^{+\infty }f\left(x,y\right)e^{j2\pi (ux+vy)}dudv.$$

Our high-frequency momentum perturbation is generated as shown below:(14)$$\rho_{hfb}=\epsilon *tanh\left(\nabla \frac{L_{sum}}{F^{\prime }\left(x,y\right)}\right).$$

We input this image into the model to get the loss of this image to get the high-frequency feature gradient, this gradient we will use as a guide to change the direction of the next gradient change. To better express what we do, we express the flow of our methods through pseudo-code, as shown in Figure 5.

## 4. Experiments

To guarantee the fairness of comparison with other methods, all weights for the loss function use common uniform metrics to compare methods.

### 4.1. Experimental Details

Datasets: Common objects in context, referred to as COCO [14], is a dataset published by Microsoft focused on image recognition. COCO is large and rich in object detection, segmentation, and captioning datasets, mainly taken from complex everyday scenes. It is used for object instances, object key points, and image captions. MS COCO is divided into 80 categories. Altogether, when we use COCO2017, YOLOR’s training set has 118,287 images. Moreover, the validation set for the attack has a total of 5000 images. The experiments are conducted on top of the validation set to ensure fairness.

Experimental environment: The experiments are run on the same CPU: Intel(R) Xeon(R) Gold 5220 CPU at 2.20 GHz and the GPU device Quadro RTX 5000 to ensure fairness.

Hyperreference settings: To compare the boosting effect of our proposed method, an identical (α, β, γ) = (0.7, 0.05, 0.3) is used to choose the hyperparameters of the loss function. This parameter is based on the default scale of YOLOR. In addition, for the selection of IoU, we use the same GIOU [36] for all our compared methods. The hyperparameters S1 and S2 are chosen to compare the superiority of our method more intuitively with parameters S1 = (ϵ = 0.15/255, i = 10) and S2 = (ϵ = 0.2/255, i = 10), respectively. At the same time, S3 = (ϵ = 0.4/255, i = 39) is the adversarial sample that we believe to be the best for the adversarial sample generated by our method with the minimum of perception, which is the adversarial sample we use as a migration attack experiment.

### 4.2. Evaluation Indicators

Disturbance Impact Index: Used for previous classification models and classification datasets, such as ImageNet [34]. Gradient-based attacks are global attacks, so the norm can reflect the perturbation limit of the attack on the image. In contrast, for the COCO datasets, the background occupies far more pixels than the target pixels. The norm constraint is to constrain each pixel as a whole, whereas in the actual attack, it is not necessary for us to attack all the pixels. Therefore, we introduced SSIM and PSNR metrics that are closer to human observation to judge the size of interference.

Structure Similarity Index Measure (SSIM) [15]: Mainly measures the similarity of images from three aspects: brightness, contrast, and structure. Due to SSIM being a perception model, it is more in line with the intuitive feeling of the human eye.

Peak signal-to-noise ratio (PSNR): This metric is an engineering term that represents the ratio of the maximum possible power of a signal to the destructive noise power that affects its representation accuracy. To measure the image quality after processing, we usually refer to the PSNR value to measure whether a processing program is satisfactory.

Performance Indicators: Regarding the evaluation metrics for object detection, we use COCO’s target recognition evaluation criteria: P is the accuracy rate, which is used to measure the percentage of correct predictions among all predictions; R is the recall rate, which is the number of all correct predictions as a percentage of the total targets; the AP metric considers both accuracy and completeness, so the area under the PR curve is used to represent a performance metric of the object detection model for this dataset. Its default IoU range is (0.5:0.95). The $AP_{50}$ metric represents the AP performance of the model if the IoU is greater than 0.5. performance; $AP_{75}$ is the AP performance for the more stringent case of IoU greater than 0.75; $AP_{S}$, $AP_{M}$, $AP_{L}$, respectively, represent the detection performance of small (area < $32^{2}$), medium ($32^{2}$ < area < $96^{2}$), and large (area > $96^{2}$) targets in the IoU range (0.5, 0.95). Table 1 shows the AP metrics of the adversarial samples we generated by YOLOR compared with the I-FGSM and PGD methods. Table 2 shows the recall performance comparison of the above methods comparison.

### 4.3. Generalizability Comparisons

As the existing object detection attack methods are only applicable in reality to evade detection, to demonstrate the effectiveness of our method more intuitively, we first reproduce the I-FGSM and PGD in a classification attack and apply it to the attack on the object detection model. To ensure fairness, all of our experiments attack both the confidence and bounding regression box of the object detection model as well as the classification results. In addition, they all use the same parameter selection of (α, β, γ) = (0.7, 0.05, 0.3). By way of comparison reference, we use the same 10 iteration times. The performance metrics are given by the authors we use in this section of experiments to compare the effectiveness of our work. To prove our conclusions more rigorously, all our experiments are built based on what has been replicated.

Attack effect experiment: From the first row of our main experiment results, Table 1, we can see that YOLOR-CSP obtains a robust AP performance of 49.2% on clean images. The second and third rows represent the performance impact of the adversarial samples generated by I-FGSM and PGD on YOLOR that we use as a reference. They both use infinite norm de-constraint with a constraint range of 8/255. It is evident that both methods generate adversarial samples with extremely high similarity, with an average SSIM close to 0.9, as well as a high peak signal-to-noise ratio. However, there is no significant impact on the YOLOR attack. As our method is an object-focused attack with no parametric to constrain the perturbation, we use the GLH method with higher S1 parameters for both SSIM and PSNR than the previous two. Compared with I-FGSM and PGD, our method GLH improves the attacks by 18.2% and 17.4%, respectively. This indicates that our attacks are not only superior to other methods but also that the perturbations generally have less impact on the images.

To achieve better attacks, we liberate the infinite parametric constraints of I-FGSM and PGD, and the performance is significantly enhanced. However, YOLOR still has 23.1% and 21.6% of AP performance. Because these methods only consider the direction of the gradient and do not quantify the magnitude of the perturbation, the attacks against the object detection model are far from satisfactory. Our method with parameter S2 is 17.7% and 12.0% more effective than I-FGSM and PGD, respectively, and our similarity is still higher than the first two.

To achieve the ultimate attack effect, we use parameter S3 to perform high-performance damage to the image while ensuring that the SSIM is not lower than 0.7. We can see that the effect of our attack makes the model lose 90% of the AP performance, basically knocking down the model’s judgment. Our later migratory experiments all use this parameter for the attack.

Recall attack experiment: In Table 2, we test the effect of the adversarial sample on the recall. Here, max = n means retaining the top n prediction boxes in confidence ranking on each graph of the test set separately. We can see that our method still outperforms I-FGSM and PGD in each metric. We can see that at max = 1, our GLH (S3) with the parameter setting almost crushes the object detection module and the recall rate drops from 37.6% to 8.8%, directly reducing the model’s recall metric by 76.5%. In addition, in the case of higher fault tolerance max = 10 and max = 100, our method discriminates to reduce the recall index of the model by 71.19% and 68.89%. In addition, for small targets, the AP metric after our attack is only 5.8%, which means that our method enables the model to ignore almost all small targets. Moreover, for medium targets, there is also only a 19.9% recall metric left, which almost loses the ability to judge. As for the recall metric for large targets, although we only reduce it to 34.9%, we also reduce the performance by 56.9% compared to a clean image, achieving an extremely significant attack effect. The reason for this is that our attacks are focused on objects, so the attack effect is especially effective for small and medium targets. For large objects, more scrambling is needed to interfere with its judgment, therefore the scrambling of the image is also increased, which also achieves a strong attack effect.

Transportability experiments: The adversarial sample we implemented through YOLOR is also highly transferable. To support our view, we chose models from recent years or more representative models for testing. Moreover, we find that our generated adversarial samples also achieve surprising results in black-box attacks, as shown in Table 3: our YOLOR-based adversarial samples also obtain quite high transferability for different backbone YOLO models. Starting from the table, we can see that for the YOLOv5, we reduce its performance from 37.40% to 15.30%, which corresponds to a performance loss of 59.09%. For YOLOX [37] and YOLOv4 [38], which have the same backbone as YOLOv5, they have a performance loss of 54.29% and 73.9%. As for the different backbone models YOLOv6 [39] and YOLOv7 [40], which are the newest and most powerful models in the YOLO family, they lose 57.30% and 68.09% of performance, respectively, for the black-box attacks we generate against the samples.

More importantly, against the non-YOLO models, our attacks also have strong migration attack performance, as shown in Table 4. For the detection performance of DETR [41] and EffcientDet [42], our adversarial samples likewise cause a high-intensity black-box attack effect on this model.

Module ablation: To ensure the effectiveness of each module, rigorous ablation experiments are conducted, as shown in Table 5. To better express the effectiveness of the work we have done, we use the GLH with S3 parameters as a sample of ablation experiments. We can see that after using our established object detection attack generalized AGA, the attack effect is especially powerful, whereas the similarity is only 0.668, as the gradient information of individual images is quite different. When we check the quality of the adversarial sample generation, most of the adversarial samples are perturbed overly severely. The samples with too severe perturbation, which have excessive initial gradients, result in perturbations that are unusual from the normal adversarial samples after iteration. The image distortion is already noticeable to our human eyes for such samples. For these adversarial samples, this perturbation is substandard.

After using the LFA module, we found that the similarity between the attacked image and the original image is improved substantially. To represent the performance of our module more intuitively, we counted the number of samples in each similarity range, as shown in Figure 6.

It is observed that in the gradient attack without adding the LFA module, there are extraordinarily many samples with the SSIM less than 0.5. In addition, such images are indistinguishable by the human eye, because the unrestricted perturbation is extraordinarily powerful for the destruction of the images. Although it obtains an amazing attack effect, we think this kind of antagonistic sample is meaningless. The adversarial sample should focus on the attack effect as well as the overall similarity distribution. as well as after adding the LFA module, we can see that the similarity of the images mostly exceeds 0.5.

IoU ablation: In the adaptive gradient attack module, we set the parameter O, which represents a criterion for measuring the accuracy of detecting the corresponding object. In addition, all our experiments have taken the value of O as GIOU. GIoU’s performance of our selected curve is better while sacrificing only the subtle similarity. As shown in Table 6. Whereas EIoU has extremely powerful performances but sacrifices too much similarity, the confrontation samples need to guarantee better similarity before we consider the performance improvement. Therefore, our parameter O takes GIoU.

Parametric ablation: For the third part of Equation (Equation 5), we set three hyperparameters α, β, and γ. We adjusted the values of each parameter separately and analyzed the effect of each parameter on the results by the experimental results, as shown in Figure 7. For the weights of confidence loss α and boundary regression box loss β, it is observed that the effect on the attack increases significantly with the increase of the parameters. However, it is also affecting the similarity of the images. At the same time, we can see that the SSIM also starts to decrease with the value of the parameters. This is explained by the fact that our parameters affect the output of this loss function and also increase the amount of perturbation, whereas the classification loss weights γ. It is apparent to us that as γ increases, although the attack effect is also enhanced, it is also obvious that the effect of this parameter on the image is enormous. For classification, more images need to be disturbed to guide the category into another class. Nevertheless, for the effect, it is considered more cost-effective to attack the confidence and bounding regression boxes.

For a more visual presentation of the functionality of the LFA module, we verified the effect of the δ parameter on the AP and tested the effect of δ from 10 to 100. As illustrated in Figure 8, the effect of the parameter on AP has been significantly reduced when δ is taken to 40, as well as the trend of AP reduction being leveled off. However, to avoid the δ parameter leveling off before 40, our main experimental values are used at 50 to avoid the effect of the parameter.

### 4.4. Visualization

To more visually represent what we have done, we used EigenCAM [47] to visualize our attack process. As shown in Figure 8, the leftmost side shows the original clean image. In addition, the detection model is used to generate a heat map by obtaining the detection results.

We can see that the model’s attention is on the salad itself, basically focusing on the detected target. Moreover, after the attack, the model’s attention to that image started to change. As the number of attacks increases, we find that after the fifth iteration, the attention of the image has deviated far from the normal value, as well as after the tenth iteration of the attack, the target has completely failed to recognize the image. Thus, this experiment proves that our attack is quite effective and lethal for the model of object detection.

## 5. Conclusions

In this work, we proposed GLH to obtain gradient information from a global perspective and focus perturbations on objects to generate adversarial samples from a local perspective. Moreover, the quality of the adversarial samples is improved by dynamic constraints and high-frequency momentum. We sufficiently demonstrate the advantages of our proposed method in white-box attacks and black-box attacks in our experiments. In addition, the adversarial samples we generate can serve as the basis for the interpretability of deep neural networks, as we destroy the model’s region of attention to that image, i.e., the learned features. Whereas adversarial attacks against object detection have been rarely studied so far, our work aims to lead the research on adversarial attacks from classification tasks to the field of object detection, as well as to promote researchers’ research on robustness against object detection and improve the application of object detection in reality.

## Figures and Tables

**Figure 1 entropy-25-00461-f001:**
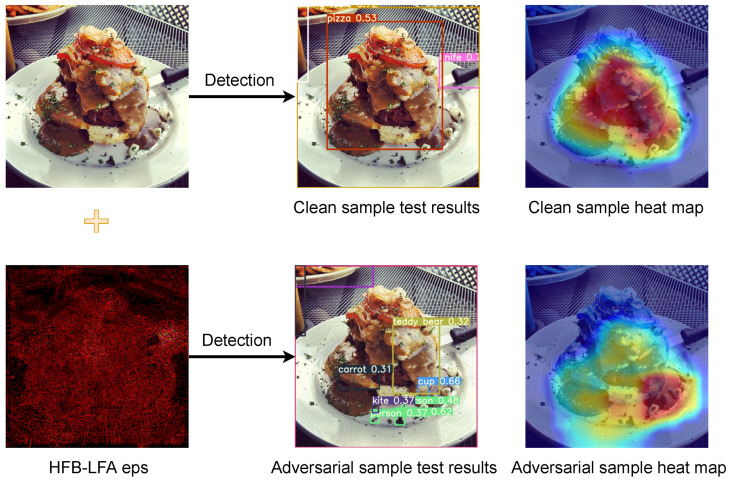
The adversarial attack against YOLOR: A clean image of the detection results in YOLOR is shown in the first row along with the model attention heat map. As we added local perturbations (we enlarged the perturbations and adjusted the color balance for visual effect), the detection results showed a huge deviation, which we can observe more visually in the heat map as the model attention changes.

**Figure 2 entropy-25-00461-f002:**
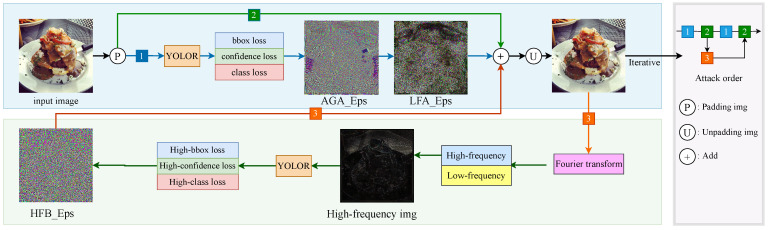
GLH: object detection against attack overall architecture.

**Figure 3 entropy-25-00461-f003:**
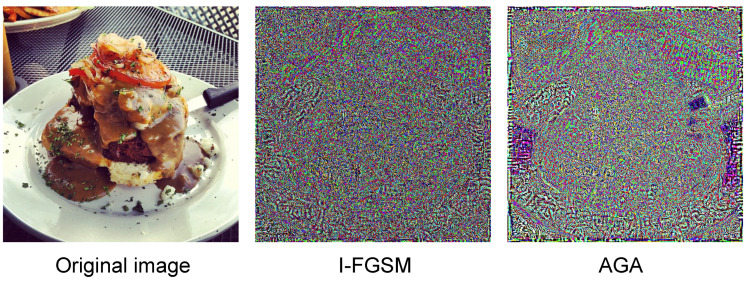
The middle perturbation image is generated by I-FGSM without the parametric constraint, whereas the one on the right is our proposed AGA perturbation generation method. Comparing the original images, we can see that the AGA perturbation generated by gradient adaptation focuses more on perturbing the region of interest of the model.

**Figure 4 entropy-25-00461-f004:**
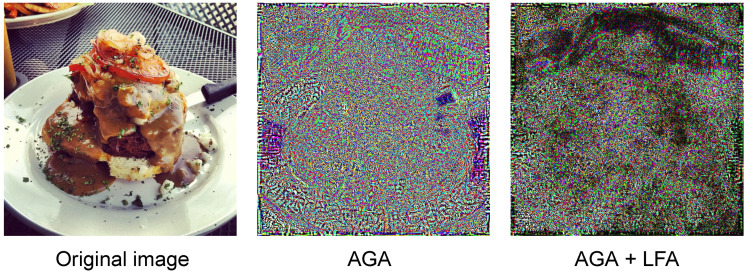
The middle image represents the perturbation generated by the AGA method, whereas the right image is the perturbation generated by adding LFA. Compared with the original image, we can see that the perturbation with LFA is more focused on the target and focuses more carefully on the area of interest of the model.

**Figure 5 entropy-25-00461-f005:**
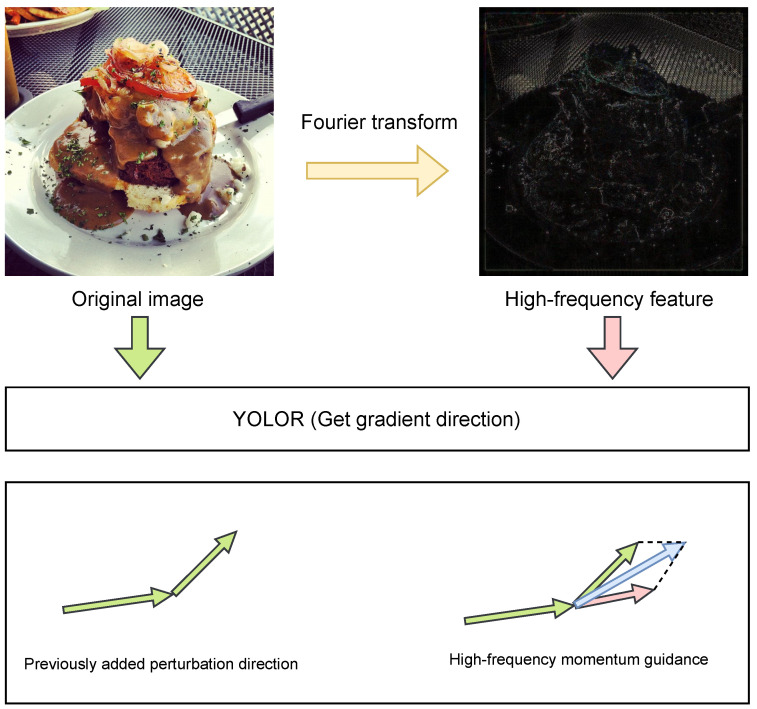
The green arrow represents the direction of gradient perturbation addition without the addition of high-frequency guidance. After adding the pink arrow, which is the high-frequency momentum guidance, the gradient attack is corrected to the blue arrow direction.

**Figure 6 entropy-25-00461-f006:**
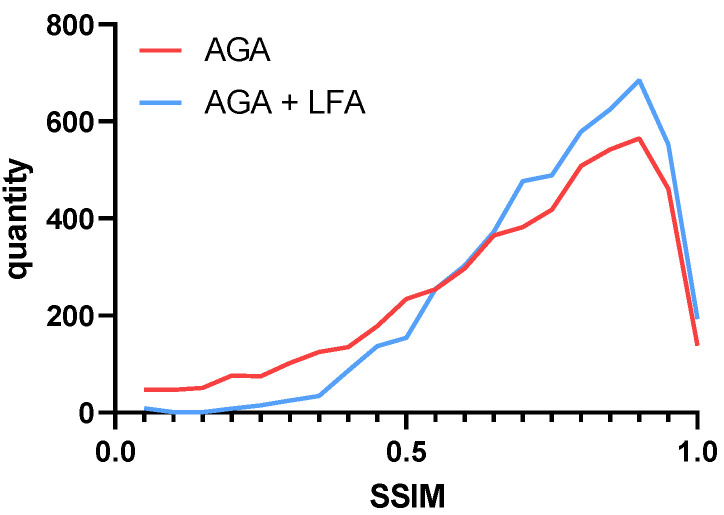
Overall similarity distribution after perturbing images.

**Figure 7 entropy-25-00461-f007:**
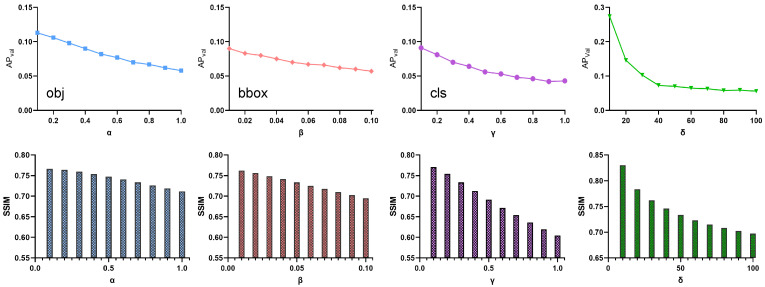
Superparametric ablation in $L_{sum}$ and δ.

**Figure 8 entropy-25-00461-f008:**
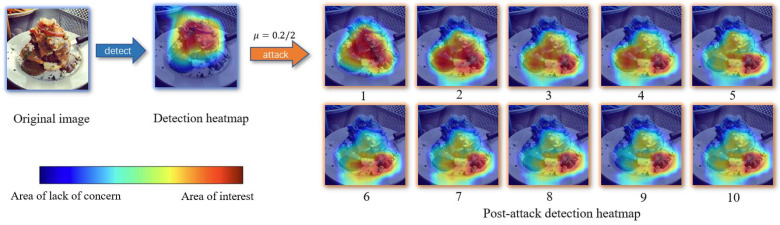
Attentional heat map based on YOLOR model. The clean image is represented on the far left, whereas the second graph shows the heat map of the clean image output under YOLOR’s object detection, with red representing the location of the model’s attention and blue representing the model’s attention to the non-focused region. The 10 pairs of graphs on the right side represent the change in the attention of the image after each attack from iteration 1 to 10.

**Table 1 entropy-25-00461-t001:** Performance of AP metrics on the COCO2017 dataset for the control sample.

Model	Constraints	Size	${\mathrm{AP}}^{\mathrm{val}}$	${\mathrm{AP}}_{50}^{\mathrm{val}}$	${\mathrm{AP}}_{75}^{\mathrm{val}}$	${\mathrm{AP}}_{S}^{\mathrm{val}}$	${\mathrm{AP}}_{M}^{\mathrm{val}}$	${\mathrm{AP}}_{L}^{\mathrm{val}}$	SSIM	PSNR
YOLOR-CSP	-	640	49.20%	67.60%	53.70%	32.90%	54.40%	63.00%	-	-
I-FGSM [17]	$L_{\infty }$	640	30.70%	45.70%	32.30%	14.40%	32.30%	46.60%	0.879	34.19
PGD [18]	$L_{\infty }$	640	30.40%	45.40%	31.90%	13.90%	31.80%	46.00%	0.881	34.24
GLH(ϵ,i)∈S1	-	640	**25.10%**	**38.60%**	**26.00%**	**9.90%**	**25.30%**	**40.70%**	**0.897**	**36.02**
I-FGSM [17]	-	640	23.10%	35.40%	24.00%	9.10%	23.70%	37.50%	0.835	31.94
PGD [18]	-	640	21.60%	33.30%	22.50%	8.00%	22.20%	35.40%	0.831	32.09
GLH(ϵ,i)∈S2	-	640	**19.00%**	**29.80%**	**19.20%**	**6.60%**	**18.70%**	**32.50%**	**0.859**	**34.03**
GLH(ϵ,i)∈S3	-	640	**4.90%**	**8.30%**	**4.80%**	**0.80%**	**3.50%**	**12.10%**	0.700	27.82

**Table 2 entropy-25-00461-t002:** Performance of AR metrics on the COCO2017 dataset for the control sample.

Method	Constraints	Size	${\mathrm{AR}}_{\max=1}^{\mathrm{val}}$	${\mathrm{AR}}_{\max=10}^{\mathrm{val}}$	${\mathrm{AR}}_{\max=100}^{\mathrm{val}}$	${\mathrm{AR}}_{S}^{\mathrm{val}}$	${\mathrm{AR}}_{M}^{\mathrm{val}}$	${\mathrm{AR}}_{L}^{\mathrm{val}}$	SSIM	PSNR
YOLOR-CSP	-	640	37.60%	61.80%	67.20%	50.80%	72.70%	81.00%	-	-
I-FGSM [17]	$L_{\infty }$	640	27.70%	47.40%	53.20%	31.50%	57.50%	71.60%	0.879	34.19
PGD [18]	$L_{\infty }$	640	27.50%	47.10%	52.90%	30.70%	57.40%	70.30%	0.881	34.24
GLH(ϵ,i)∈S1	-	640	**24.00%**	**42.40%**	**48.00%**	**26.30%**	**51.50%**	**66.60%**	**0.897**	**36.02**
I-FGSM [17]	-	640	23.60%	41.60%	46.90%	25.20%	50.60%	65.30%	0.835	31.94
PGD [18]	-	640	22.70%	40.10%	45.70%	24.30%	49.10%	63.70%	0.831	32.09
GLH(ϵ,i)∈S2	-	640	**20.30%**	**36.70%**	**42.00%**	**21.10%**	**44.80%**	**60.30%**	**0.859**	**34.03**
GLH(ϵ,i)∈S3	-	640	**8.80%**	**17.80%**	**20.90%**	**5.80%**	**19.90%**	**34.90%**	0.700	27.82

**Table 3 entropy-25-00461-t003:** Migration attacks for the YOLO family.

	YOLOR-CSP [33]	YOLOv4-pacsp-s [38]	YOLOv5-s	YOLOX-s [37]	YOLOv6-s [39]	YOLOv7 [40]
Backbone	DarkNet53	DarkNet53	DarkNet53	DarkNet53	EfficientRep	ELANNet
Base	49.20%	38.90%	37.40%	39.60%	43.80%	51.40%
YOLOR-CSP	4.90%	10.30%	15.30%	18.10%	18.70%	16.40%

**Table 4 entropy-25-00461-t004:** Migration attacks on other models.

	DETR [41]	Efficientdet-d5 [42]	Mask R-CNN [43]
Backbone	Resnet50	EfficientNet	Resnet50
Base	42.00%	50.00%	30.90%
YOLOR-CSP	15.70%	22.00%	8.50%

**Table 5 entropy-25-00461-t005:** Module ablation experiment.

	LFA	HFB	AP	SSIM	PSNR
Clean			0.492	-	-
AGA			**0.053**	0.668	24.81
AGA	✓		0.071	**0.733**	**29.08**
AGA		✓	0.053	0.668	24.80
AGA	✓	✓	**0.070**	**0.733**	**29.08**

Symbol ✓ Represents the addition of the module.

**Table 6 entropy-25-00461-t006:** IoU ablation experiment.

Model	IoU	AP	SSIM	PSNR
YOLOR-CSP	EIoU [44]	6.10%	0.699	28.05
YOLOR-CSP	DIoU [45]	7.30%	0.734	29.14
YOLOR-CSP	CIoU [46]	7.20%	0.735	29.14
YOLOR-CSP	GIoU [36]	7.10%	0.733	29.09

## Data Availability

Not applicable.

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
