# Peer review of "GLH: From Global to Local Gradient Attacks with High-Frequency Momentum Guidance for Object Detection"

_entropy, 2023, doi:10.3390/e25030461_

Round 1

Reviewer 1 Report

This paper reveals the security problem of deep learning models to generate adversarial samples from another perspective with a high degree of migration.

After reviewing your paper carefully, here are some suggestions:

a) In the first section, the background introduction at the beginning is not concise enough and is rather redundant. Please refine that section.

b) The flowchart symbols in Section 3 are not clear enough, and I would like to see a more detailed representation of your work.

c) For the experimental data, we would like to have more details about the experimental background or the experimental environment in the header of the table.

d) There are symbolic errors below Equation 9 and some grammatical or punctuation errors in the text, please correct them carefully.

e) In the experimentation part, I consider that more models are needed to show your ability to migrate attacks.

Reviewer 2 Report

This paper proposes a method called From Global to Local Gradient Attacks with Highfrequency Momentum Guidance for object detection. I have several concerns about this paper:

(1) Its seems only COCO dataset is used in experiment. More large datasets should be used for evaluation.

(2) The language should be largely improved. 

(3) Some related works are missing, such as "An Optimized Black-Box Adversarial Simulator Attack Based on Meta-Learning. Entropy 24(10): 1377 (2022)", and "Simulator Attack+ for Black-Box Adversarial Attack. ICIP 2022: 636-640". 

Reviewer 3 Report

The author should be congratulated for conducting an interesting article. Overall, the manuscript is clear and concise manuscript. Please find my comments in the review file

Round 2

Reviewer 1 Report

The paper looks good and the concerns are addressed well.

Reviewer 2 Report

The authors have solved my concerns. I have no additional concerns about this paper.

Reviewer 3 Report

The authors revised the manuscript in light of my comments and provided reasonable responses. I consider the manuscript to be well written and
consistent with the scope of the journal's research and recommend its publication.